# Focus on PD-1/PD-L1 as a Therapeutic Target in Ovarian Cancer

**DOI:** 10.3390/ijms232012067

**Published:** 2022-10-11

**Authors:** Adrian Dumitru, Elena-Codruta Dobrica, Adina Croitoru, Sanda Maria Cretoiu, Bogdan Severus Gaspar

**Affiliations:** 1Department of Pathology, Carol Davila University of Medicine and Pharmacy, 050474 Bucharest, Romania; 2Department of Pathology, University Emergency Hospital of Bucharest, 050098 Bucharest, Romania; 3Department of Pathophysiology, University of Medicine and Pharmacy of Craiova, 200349 Craiova, Romania; 4Department of Dermatology, Elias University Hospital, 011461 Bucharest, Romania; 5Department of Medical Oncology, Fundeni Clinical Institute, 022328 Bucharest, Romania; 6Department of Oncology, Titu Maiorescu University, 031593 Bucharest, Romania; 7Department of Cell and Molecular Biology and Histology, Carol Davila University of Medicine and Pharmacy, 050474 Bucharest, Romania; 8Surgery Department, Carol Davila University of Medicine and Pharmacy, 050474 Bucharest, Romania; 9Surgery Clinic, Bucharest Emergency Clinical Hospital, 014461 Bucharest, Romania

**Keywords:** PD-1/PD-L1, ovarian cancer, gynecological malignancy, miRNAs, lncRNAs

## Abstract

Ovarian cancer is considered one of the most aggressive and deadliest gynecological malignancies worldwide. Unfortunately, the therapeutic methods that are considered the gold standard at this moment are associated with frequent recurrences. Survival in ovarian cancer is associated with the presence of a high number of intra tumor infiltrating lymphocytes (TILs). Therefore, immunomodulation is considered to have an important role in cancer treatment, and immune checkpoint inhibitors may be useful for restoring T cell-mediated antitumor immunity. However, the data presented in the literature until now are not sufficient to allow for the identification and selection of patients who really respond to immunotherapy among those with ovarian cancer. Although there are some studies with favorable results, more prospective trials are needed in this sense. This review focuses on the current and future perspectives of PD-1/L1 blockade in ovarian cancer and analyzes the most important immune checkpoint inhibitors used, with the aim of achieving optimal clinical outcomes. Future studies and trials are needed to maximize the efficacy of immune checkpoint blockade therapy in ovarian cancer, as well as in all cancers, in general.

## 1. Introduction

Malignant gynecological pathology is far dominated by ovarian cancer, which causes the greatest number of deaths (Figure 1). Ovarian cancer is one of the deadliest cancers and considered, after cervical and uterine cancers, the third most frequent type of gynecological cancer [1]. Having a lower frequency than breast cancer, ovarian cancer has a worse prognosis because it is three times more lethal [2].

Globally, it is anticipated that the mortality rate of ovarian cancer will increase, following the trend of the general cancer burden, which is expected to rise 47% by 2040 [1,5]. The high mortality rate of patients with ovarian cancer is due to the presence of non-specific symptoms or complete absence of symptoms in the early stages of disease [6]. Detection in early stages is also made difficult by the absence of early screening programs and specific diagnostic markers [7,8]. A total of 90% of ovarian cancers are caught in stages I and II and, according to the FIGO (International Federation of Gynecologists and Obstetricians) classification, have a 5-year survival rate of 70%, whilst, in late stages (III and IV, according to FIGO), the survival rate decreases to 30%. Due to the relapses that frequently occur in approximately 60% of ovarian cancer patients in remission, it is necessary to constantly evaluate new treatment methods that involve anti-angiogenic factors, targeted immunotherapy, poly-DNA-ribose polymerase inhibitors, or their combinations with classical methods, such as would-be chemotherapeutic agents [6].

The last 20 years have been marked by accelerating cancer research as a global priority. Therefore, there is a need for a permanent review of the literature on the ever-changing aspects of ovarian cancer histopathology, as well as the latest molecular therapies associated with it. Establishing a correct treatment depends on multiple factors, including the tumor grading (G1–G3) and the histological variety of ovarian cancer [9]. While for almost 80% of the patients, primary treatment based on surgery and platinum-based chemotherapy has a favorable prognostic, for others, new therapeutic strategies have to be taken into consideration [10,11]. Although surgical resection, followed by staging, is the most efficient treatment, neoadjuvant therapy is often required in patients with advanced disease, usually with a poor prognosis for long-term survival. Patients who undergo complete resections of visible tumors and receive IV/IP chemotherapy have been shown to have the most consistent and best results and outcomes [12].

The vast majority of the ovarian tumors are of epithelial origin and classified in two subtypes, types I and II, with prognostic and predictive value. Type I, representing approximately 30% of ovarian cancers, is typically genetically stable and slow-growing, and it comprises low-grade serous, endometrioid, clear-cell, and mucinous carcinomas. Type II represents 70% of all ovarian cancers and is more aggressive, genetically unstable, and includes high-grade serous carcinomas. This classification can partially explain the capacity of tumoral cells to escape immune clearance, resulting in the propagation of cancer [12].

Nowadays, immune checkpoint therapy has reached the goal of being considered very useful, in combination with surgery, chemotherapy, radiotherapy, and other treatments, in the struggle to cure cancer and raise the survival rate [13]. However, most ovarian cancers evade the host immune system, due to an unbalance between immune tolerance and immune response. Around twenty years of research generated a vast description of the cancer pathogenesis and its self-sustainability mechanisms that led to the possibility of applying a personal treatment scheme to a specific category of patients [14]. The quantum leap in understanding how antitumor immunotherapy functions was represented by the discovery of the directives that rule the CD4+ and CD8+ lymphocytes immune response. This unraveled a tremendous cascade of immunological checkpoints that obviate improper activation phenomenon or supplement excitatory stimuli of the immune response [15]. The immunological escape of the tumor could be mediated by many pathways, i.e., PD-1/PD-L1 or PD-L2, CTLA-4, IDO, NO2, ARG-1, VEGF, and PGE2 [16]. For example, the expression of CTLA-4 and PD-1 on the lymphocyte membrane is responsible for the regulation of the T cell response [14]. One relevant example is the use of antibodies against programmed death receptor (PD-1), a molecule expressed on the surface of several immune cells, including macrophages and dendritic cells, as well as on the surface of mesenchymal and activated vascular endothelial cells [17]. This receptor was identified and studied by Ishida et al., who were trying to discover a gene that induces programmed cell death [18]. One notable milestone in the immunotherapy of cancer, in general, as well as for ovarian cancer, in particular, was touched on when studies on the programmed cell death receptor 1 (PD-1) and its ligands programmed death-ligand 1 and 2 (PD-L1/PD-L2) interactions were revealed [19]. In 2007, PD-Ls were shown to be expressed by ovarian tumors and to induce T cell anergy or apoptosis, favoring ‘‘tumor immune escape’’, and they were proposed as prognostic factors of human ovarian cancer [20].

The aim of this review is to emphasize the importance of this particular area of immunotherapy, based on the PD-1/PD-L1 interaction for the malignant ovarian tumors. Moreover, it is important to raise awareness, regarding the existence of a variety of treatments and means by which they can be prescribed in direct correlation with clinical and paraclinical data. The newer immune agents can be used via the combination of original chemotherapy or alone, based on their target sites.

## 2. PD-1, Its Ligands and Evading Anti-Tumor Response

Cancerous cells are usually eliminated by the immune system. The proliferating masses of tumoral cells are invaded by tumor-infiltrating lymphocytes (TILs), which is considered to be a favorable prognostic factor in many cancers, e.g., colorectal, esophageal, melanoma, and ovarian cancers, to name but a few [21]. Depending on the density of TILs, ovarian tumors were classified as “hot” or “cold” tumors [22]. T lymphocytes play a central role in the adaptive immune system, through its capacity for antigen-directed cytotoxicity (for detailed review see [23]). Two or more distinct signals are required by T cells, in order to become fully activated during antigen presentation, whereas the first signal required is antigen-specific, thus being delivered by the T cell receptor (TCR) upon its interaction with the antigenic peptides that are presented by the antigen-presenting cells (APC), which exhibit the major histocompatibility complex (MHC) molecules on their surfaces; the second signals are antigen-independent and commonly provided by the interactions between ligands on antigen presenting cells and specific co-receptors on T cells [24]. These co-receptors are generally classified in two types: stimulatory and inhibitory [24]. Stimulatory receptors, such as CD28, can be countered by the inhibitory ones, such as cytotoxic T lymphocyte antigen 4 (CTLA-4), the latter being involved in the inhibition of T cell functions, therefore blocking the activation [25]. One of the most important inhibitory co-receptors expressed by T cells is the programmed death-1 (PD-1) receptor, which is able to play critical roles in cancer immunology [26]. T cells may play a role in the new era of cell and gene therapy in solid tumors because the chimeric antigen receptor (CAR) technology has been continuously developed. Despite its approval, since 2017, for hematological malignancies, where CAR-T therapy has been successful used in the treatment of solid tumors (including ovarian cancer), the adverse effects and low response rate are still important issues to address [27,28]. Trials of CAR-T cells in ovarian cancers are ongoing (NCT03585764; NCT02498912).

Programmed cell death receptor 1 (PD-1) is encoded by the Pdcd1 gene on chromosome 2 in humans (in mice, this gene is located on chromosome 1), a gene that consists of 5 exons, with exon 1 encoding a short signal sequence and exon 2 encoding the Ig domain [29]. Exon 3 is responsible for encoding a ~20 amino acid (aa) stalk and the transmembrane domain, whereas exon 4 codes for a short 12 aa sequence that represents the beginning of the cytoplasmic domain [29]. Exon 5 contains the C-terminal intracellular residues and a long 3′UTR [25]. PD-1 is a 50–55 kDa, 288 aa transmembrane glycoprotein that lacks the membrane-proximal cysteine residue required for homodimerization that is present in the structure of the other members of the CD28 family, thus rendering PD-1 monomeric, both in solution and on cell surface, as structural and biochemical analyses show [26]. The cytoplasmic domain presents two tyrosine residues, with the membrane-proximal one constituting an immunoreceptor tyrosine-based inhibitory motif (ITIM) and the second one constituting an immunoreceptor tyrosine-based switch motif (ITSM) [30].

The ligands of PD-1, PD-L1 (B7-H1; CD274), and PD-L2 (B7-DC; CD273) are type I transmembrane glycoproteins consisting of IgC- and IgV-type extracellular domains, with the latter presenting a higher affinity for PD-1 than the former, but it is expressed by significantly fewer cell types. PD-L2 is not commonly present on resting cells, but can be inducible expressed on dendritic cells, macrophages, cultured bone marrow-derived mast cells, and some subsets of B lineage cells [31,32]. On the other hand, PD-L1 is expressed on both hematopoietic and non-hematopoietic cells; it is expressed at a high basal level on B cells, dendritic cells, macrophages, and mast cells, while being further up-regulated when activated [33]. The greater part of the non-hematopoietic cells that express PD-L1 originate from solid malignancies, such as ovarian carcinoma, renal cell carcinoma, and non-small cell lung cancer. A very important aspect is that increased aggressiveness and risk of death have been linked with high tumor expression of PD-L1 [33].

PD-L1 is a 290 aa type I transmembrane glycoprotein encoded by the Cd274 gene on human chromosome 9, a gene that consists of seven exons, the first of which is non-coding and contains the 5′UTR. Exons 2, 3, and 4 contain a signal sequence, i.e., the IgV-like and IgC-like domains. The next two exons, 5 and 6, encode the transmembrane and intracellular domains, whereas the last exon, exon 7, codes for intracellular domain residues and the 3′UTR. The intracellular domain of PD-L1 is short (about 30 aa) and highly conserved in all reported species [29].

PD-L2 is also a type I transmembrane glycoprotein encoded by the Pdcd1lg2 gene, which is adjacent to Cd274 gene, being separated by only 42 kb of intervening genomic DNA in humans. This gene also consists of 7 exons, the first of which is non-coding; the second one codes for the signal sequence. The third exon encodes the IgV-like domain, the fourth one encodes the IgC-like domain, and the fifth contains a short stalk, the transmembrane region, and the beginning of the cytoplasmic domain. In humans, the last two exons encode for a cytoplasmic domain of 30 aa, which is longer, compared to the cytoplasmic domain of other species, such as mice [29].

The PD-L1 and PD-L2 transcripts are expressed by a wide range of normal tissues, with high expression levels being recorded in the placenta, lung, ovaries, heart, and liver, and low levels of expression in the thymus, lymph nodes, and spleen; the brain shows an absence of their expression [34,35,36,37]. The increased expression of PD-1/PD-L-1 on tumor and immune system cells in patients with ovarian cancer has been demonstrated over time [38]. The interaction between PD-1 and its ligands limit the inflammatory response by inhibiting the function of cytotoxic T cells (Figure 2) [39]. The overexpression of PD-1 and its ligands leads to decreased immune response and tolerance of the neoplastic process [40]. Thus, inhibiting the activity of PD-1/PD-L 1 may be very effective. The evaluation of PD-1/PD-L 1 in the specimen samples from patients with ovarian tumors is realized by immunostaining. PD-1 and PD-L1 are expressed by various solid tumors and their subtypes, with ovarian tumors presenting PD-1 expression in 93% of studied cases, whereas PD-L1 is expressed in only 43% of cases. When compiling data, concurrent PD-1 and PD-L1 expression was identified in 36% of the studied cases. Additional testing revealed that ovarian cancers were frequently (70–100%) infiltrated with PD-1+ tumor-infiltrating lymphocytes (TILs). When present, PD-1+ TILs density varied from 1 to >20/hpf [41].

## 3. Various Cancer Types and the Linking between microRNAs, lncRNAs and PD-1/PD-L1

MicroRNAs consist of about 22 nucleotide, single-stranded, non-coding RNA molecules that regulate gene expression at the post-transcriptional level [42]. The expression of different miRNAs is dysregulated in cancer at the genetic or epigenetic levels or at the level of biogenesis or the recruitment of transcription factors. In malignant cells, microRNAs levels are altered by gene amplification, deletion or translocation (leading to increased cell death resistance), modified signaling pathways, and diverse mechanisms responsible for high invasiveness and progression [43].

In non-small cell lung carcinoma, PD-1/PD-L1 expression is downregulated through microRNA-138 at the level of dendritic cells and T lymphocytes. MicroRNA-138 is also responsible for the down-regulation of the proliferation capacity of the tumoral cells and by increasing the number of tumor-infiltrating dendritic cells [44].

In lung adenocarcinoma, miR-33a can inhibit lung tumor growth, cell proliferation, and cell cycle progression. In this sense, an inverse correlation between miR-33a and PD-1 expression and levels was described [45]. Patients with low-grade tumors and female gender were detected with high levels of miR-33a and low levels of PD-1, and they were reported to have a better prognosis, thus suggesting that microRNA-33a could be used as a prognostic marker in the future [46].

PD-1 expression in hepatitis B virus (HBV)-associated liver diseases was studied by Zhang et al., who demonstrated that there is a decrease of PD-1 expression in lymphocytes and microRNA-4717 levels were significantly decreased in the T lymphocytes, while IFN-γ and TNF-α levels were increased in patients with chronic hepatitis and hepatocellular carcinoma post-HVB infection (rs 10204525, genotype GG) [47].

In the metastatic clear-cell carcinoma of the kidney, Incorvaia et al. described a subset of silenced microRNA, called a “lymphocyte miRNA signature”. This set was found to be overexpressed in patients who had a favorable response to nivolumab therapy, which was probably acting on the expression of microRNA-22. Moreover, the same study found an attenuated tumor immune response, which was observed in patients with low levels of microRNA-339 and high levels of PD-L1 [48].

Anti-PD-1 and -PD-L1 antibodies can be targeted in lymphoma cells by MiR-155, a significant player in the promotion of tumor immune escape. MiR-155 is also involved in the tumoral growth, through the inhibition of tumor associated immune cells and triggering apoptosis in CD8-positive T lymphocytes [45].

Long non-coding RNAs (lncRNAs) have no protein-coding capacity and are very similar in structure to mRNA, consisting of more than 200 nuclotides. These non-coding RNAs are involved in gene expression regulation: epigenetic regulation, transcriptional regulation, and post-transcriptional regulation [49]. LncRNAs are involved in many cellular processes, such as cell differentiation, cell cycle regulation, and even cancer development, with their misexpression creating conditions for tumor initiation, growth, and metastasis [50,51,52].

Long non-coding RNAs (lncRNA) are as also involved in metabolic balance and modulation of the immune response [53]. An increasing number of lncRNA were demonstrated to be oncogenes or tumor suppressors [54]. Metastasis-associated lung adenocarcinoma transcript 1 (MALAT 1) is a lncRNA induced by transforming growth factor beta (TGF-beta), which is involved in the metastatic process in numerous cancer types, including large, diffused B cell lymphoma [55]. Throughout microRNA-195 interactions, MALAT 1 can induce the upregulation of PD-L1 and apoptosis of T cells, with the subsequent evasion of tumoral cells from the immune system [56]. Another lncRNA involved in up-regulating PD-L1 is the small nucleolar RNA host gene 14 (SNHG14) via upregulation of the zinc finger E-box binding homebox 1 [3].

In oral squamous cancer, IFITM4P lncRNA overexpression results in an excess phosphorylation of TAK1 (Thr187) and, subsequently, nuclear factor κB (Ser536), which leads to the upregulation of PD-L1 and facilitation of the escape of cancer cells via the control mechanisms of the immune system, thus favoring tumor progression [57].

Another cancer in which lncRNA plays an important role in modulating PD-L1 expression is lung cancer, through NKX2-1-AS1 (by decreasing PD-L1 expression) and NKX2-1 (by increasing its expression). The molecular mechanisms of these processes can be used to counteract the evasion of tumor cells by the immune system through the overexpression of PD-L1. The NKX2-1 gene, also known as TTF1, is a gene that plays a key role in the occurrence of lung and thyroid tumors. The expression of this gene can be quantified through empirical immunohistochemical tests; most of the time, it is of real help to pathologists to document the pulmonary or thyroid origin of a tumor. [58].

In ovarian cancer, different cells can express PD-L1: B and T lymphocytes, macrophages, tumor cells, and even dendritic cells isolated from the primary tumor and locoregional lymph nodes. PD-L1 inhibition in these patients leads to an increase in miR-424 (322) and is associated with a favorable long-term prognosis, whilst the opposite situation (increased PD-L1 and decreased expression of miR-424 (322)) is frequently related to a resistance to chemotherapy [59]. Recently, PD-L1 levels were demonstrated at high levels in the plasma of patients with ovarian cancer, compared with healthy women. The study also indicates that the low expression of miR34a correlates with low serum PD-L1 levels, and high miR200 levels correlates with high serum PD-L1 levels, thus suggesting their importance as biomarkers for anti PD-L1 immune therapy [60].

## 4. PD-L1 Targeted Immunotherapy in Cancers

Currently, there is an increased number of studies regarding the use of PD-1/PD-L1 pathway inhibitors. Malignant epithelial tumors are the most common neoplasm of the ovary, accounting for 90% of all cases of ovarian cancers [4]. Several molecular studies have been carried out, in order to determine the pathogenesis of these malignant lesions, thus revealing several aspects that subdivided the tumors into two subtypes. Type I tumors are derived from benign, extra-ovarian precursors [61]. The group includes clear-cell and endometrioid carcinomas, both of which are associated with underlining endometriosis. Mucinous carcinoma, low-grade serous ovarian carcinoma, and Brenner tumors are also included in these groups, although they are less common [62]. They tend to present as a unilateral cystic mass. Type II malignant neoplasm develops from intraepithelial lesions in the fallopian tube. High-grade ovarian carcinoma, carcinosarcoma, and undifferentiated carcinoma are considered type II lesions and characterized by a highly aggressive clinical behavior and poor prognosis. The patients are frequently discovered with advanced disease at the moment of diagnosis [63].

The expression of PD-1 on the surface of many cell types, and especially on tumors, makes it possible, through the interaction between PD-1 and PD-L1, though difficult, to get rid of cancer cells by the mechanisms of the immune system, which leads to the emergence of therapeutic resistance [14,31,64,65]. Moreover, there was an increased therapeutic resistance in certain histological types that were initially associated with a poor prognosis, but in which case immunotherapy brought an optimistic perspective [25]. Immune checkpoint inhibitors were successfully used as a battleline treatment in melanomas, non-small cell lung cancer, renal or urinary tract cancers, and breast, head, and neck cancers; they were recently introduced as an option in malignant hematological proliferations [66]. The pioneer therapy in checkpoint immune inhibitors is represented by ipilimumab, a monoclonal antibody that blocks the CTLA-4 molecule [67,68]. Ever since, novel molecules have been developed that obstruct the perpetuation of other inflammatory cascades, such as the ones provoked by the PD-1 and PDL-1 complexes. For example, nivolumab and pembrolizumab are anti-PD-1 antibodies that presented a successful rate in treating patients who were suffering from metastatic melanoma, as well as cemiplimab, from the same drug class, and they were utilized in cutaneous squamous cell carcinoma (Figure 3) [67,68,69,70]. Other examples of checkpoint inhibitors that interfere with PDL-1 incorporation are durvalumab, atezolizumab, and avelumab, which support the usual treatment protocol, especially in urothelial or non-small lung cancers [64,66,70].

## 5. Efficacy of PD-1/PD-L1 Inhibitors in Ovarian Cancer 

Despite numerous therapeutic protocols, ovarian cancer remains a major cause of neoplastic mortality [38,71]. Therefore, there are now several trials conducted that introduce the latest class of checkpoint immune inhibitors that target PD-1/PDL-1 complexes into the therapeutic protocol of FIGO stage III or IV or relapsed ovarian cancer. Understanding the programmed cell death pathway associated with the PD-1 receptor allowed for the design of clinical trials for treating ovarian cancer using antibodies against the PD-1 receptors (nivolumab and pembrolizumab) and PD-L1 ligands (avelumab, BMS-936559, durvalumab, and atezolizumab).

The efficacy of PD-1/PD-L1 inhibitors, as promising immunotherapeutic agents, was analyzed by Chen et al. in a meta-analysis based on 91 published clinical trials, phases I to III, in different cancer types, including gynecological cancers. This meta-analysis revealed that PD-1/PD-L1 inhibitors, combined with chemotherapy, had a statistically significantly higher objective response rate, compared with immunotherapy alone, and the duration of response was significantly reduced in the case of the same combination [72].

There are many factors that determine how a tumor will respond to this type of treatment. For example, pembrolizumab, not approved by FDA for ovarian cancer, showed a modest result for immunotherapy in stage II clinical trials [38,73]. This can be explained by the fact that efficiency is dependent on the presence of microsatellite instability [74,75]. In certain circumstances, pembrolizumab could be used for patients with MSI-H (microsatellite instability-high), dMMR (mismatch repair-deficient), solid tumors, and no satisfactory alternative treatment options. This was indicated in a study by Le et al., which was performed to evaluate the efficacy of the PD-1 blockade across 12 different tumor types, with MSI-H and dMMR, which were found to be sensitive to immune checkpoint blockade [76]. The incidence of dMMR in ovarian cancer is reported to be between 2–29%, with the higher frequencies being associated with younger ages of diagnosis and the presence of an increased tumoral lymphocytic infiltrate [77].

Another predictive biomarker of the efficiency of anti-PDL-1 antibodies is the overexpression of PDL-1 on the cell membranes of the malignant cells, as suggested in a study published by Chin et al. [64,65,66]. They showed that FIGO stage III and IV ovarian tumors, with a raised immune reactivity molecular subtype, expressed a high quantity of PDL-1 on the surface of tumoral cells. Understanding the fact that not all patients who undergo anti-PDL-1 checkpoint inhibitor and present an upregulation of PDL-1 on tumor cells and infiltrated inflammatory cells have a good response to therapy, they raised the dilemma of the necessity of other molecular biomarkers for an improved description of the molecular pattern of the high-grade ovarian tumors [38]. Moreover, it has been displayed in several preclinical trials that the favorable outcome from checkpoint inhibitors therapies is strongly connected to the perfect timing of initiating the treatment. It is considered that the ideal moment to administer the therapy is when the tumoral contingent reaches the maximum level of intraepithelial tumor-infiltrating lymphocytes (TILs) [74]. It has been demonstrated that PDL-1 overexpression itself downregulates the T cytotoxic CD8+ lymphocytes action and, therefore, decreases the protection of the host immune system against tumor overgrowth [78].

However, as remarkable as the response after checkpoint immune inhibitors treatment in melanoma may be, the trials results for urothelial and non-small lung cancers in ovarian tumors show a modest improvement of the median survival rate of 10–15%, until nowadays, with disease control observed in approximately 50% of patients. [78,79,80,81,82,83]. One possibility for enhancing the response to anti-PDL-1 antibodies in these types of cancers is to associate them with radiotherapy, which is known to boost the effect of anti-PDL-1 checkpoint inhibitors in other malignancies [14]. The JAVELINE 200 trial, a phase III clinical study, included 566 patients who were platinum resistant or had refractory ovarian cancer. The study compared avelumab monotherapy with the avelumab-pegylated liposomal doxorubicin combination. A therapeutic advantage was observed in patients who were treated with the combination and had PDL-1 expression on their tumoral samples [78,84]. The study presented, as main objectives, the evaluation of progression-free and overall survival; as secondary objectives, the study presents the objective response and its duration, as well as aspects related to the tolerance of the therapy and pharmacokinetics of the drug. At the moment, several clinical trials have been conducted on small-scale populations of patients for ovarian cancer, and they used the following molecules in their research: ipilimumab, tremelimumab, nivolumab, pembrolizumab, and other anti-PDL-1 antibodies, such as BMS-936559, MEDI4736, MPDL33280A, and MSB0010718C (avelumab) [85].

The first study aimed at the efficacy of PD-1 inhibitors was published by Hamanishi et al. and aimed at evaluating the role of nivolumab (a fully humanized monoclonal antibody against the PD-1 receptor) in ovarian cancer that is resistant to treatment with platinum agents, as other previously suggested [20,86,87]. The same team demonstrated the association between the high level of PD-L1 and reduced survival rate, as well as the low number of CD(+) cytotoxic lymphocytes [16].

The findings published by Webb et al., which found PD-L1 expression mainly in CD68+ macrophages, approached the topic in a different light [88]. They quantified the expression of PDL-1 in different subtypes of ovarian cancers, as well as the relationship between the expression of PDL-1 in tumor cells and the marker in immune cells associated with tumor proliferation. The authors also verified the prognostic significance of this expression, placing a special emphasis on PDL-1 expression in high-grade ovarian serous carcinomas. Their study revealed a positive relationship between PD-L1 expression and the presence of tumor-infiltrating regulatory T cells and/or lymphocytes with the CD8+ CD103+ PD-1+ phenotype. The level of PD-L1 expression in the different histopathological subtypes of ovarian cancer varied greatly with highest expression of PD-L1 in serous ovarian carcinoma (57.4%), followed by mucinous ovarian carcinoma (26.7%) and endometrioid ovarian carcinoma (24%) [88]. Clear-cell ovarian carcinoma was associated with the lowest level of PD-L1 (16.2%) [88]. The authors also demonstrated a positive correlation between PD-L1 expression and the overall survival of patients with high-grade ovarian serous carcinoma (HGSC), thus endorsing the concept of adaptive resistance of tumor cells [88]. At first glance, the studies performed by Hamanishi and Webb seem to be contradictory, and the major differences in PDL-1 expression reported by the authors seem to be related to the clones used in the immunohistochemical studies, as suggested by Webb et al. For this reason, the evaluation of PDL-1 expression in tumor cells from ovarian carcinomas (regardless of type), as well as from immune cells associated with tumors, seems to require an appropriate protocol and reporting system, similar to that used for lung carcinomas, for better traceability of results. In this sense, in addition to the classic detection of PDL-1 expression through immunohistochemical tests, alternative, innovative approaches are probably needed that could detect the presence of PDL-1 in much smaller amounts of tissue or biological fluids. Such a method was validated by Grel et al. [89]. Some studies suggest a positive correlation between the TILs and survival rate [90]. Others have underlined a relationship between high PD-L1 expression in the peritoneal fluid of these patients, as well as the formation of metastases within the peritoneal cavity, knowing that malignant serous ovarian tumors have a special predisposition for the invasion of the peritoneum [91].

Pembrolizumab is a humanized anti-PD-1 monoclonal antibody that blocks the PD-1 on the cell surface, thus preventing the PD-1/PD-L1 interaction. A phase Ib study developed by Varga et al. evaluated the role of pembrolizumab in 26 patients with advanced ovarian cancer, fallopian tube cancer, and peritoneal invasion who did not respond to first-line oncological treatment. [92]. The results of the treatment seem to be modest, since the overall response rate was only 11.5% and disease control rate was 34.6%. Complete response to treatment was obtained in only one patient, partial response occurred in two patients, and disease stabilization was confirmed in six patients [92,93,94]. A similar study using pembrolizumab as a single therapeutic agent for patients with recurrent ovarian carcinoma demonstrated a similar, modest therapeutic response [73].

In addition to pembrolizumab, other therapeutic agents have been studied, in order to diversify the immunotherapy-type therapeutic options for ovarian cancer patients. Avelumab is a fully humanized anti-PD-L1 IgG1 monoclonal antibody that inhibits the PD-L1 interaction with the PD-1 receptor [94]. The study conducted with avelumab is one of the largest studies to date involving the programmed death pathway [95]. The overall objective responses, in a group of 124 women with recurrent or refractory ovarian cancer, was 9.7%, and the disease control rate was noted in about half of the patients. PD-1 expression was assessed in 74 cases, and 57 women (77%) showed increased PD-L1 expression. However, it seems that the results of therapy with avelumab are quite modest and comparable to those with pembrolizumab; the objective response rate was 12.3%, while, in the group of women without excessive PD-L1 expression, it was 5.9% [95]. Contrary to expectations that anti-PD-L1/PD-1 therapies could be effective in ovarian cancers, clinical trials showed that their performance remains restricted—first of all, because the expression of PDL-1 is modest in ovarian carcinomas [96].

Another monoclonal anti-PD-L1 antibody, BMS-936559, was studied. This study included 17 patients with ovarian cancer, among which, 5.9% responded partially to treatment, and 17.6% achieved disease stabilization. All patients received doses of 10 mg/kg. There are also studies on the effectiveness of durvalumab—a monoclonal antibody directed against the PD-L1 protein [94].

Despite the promising results in the treatment of solid tumors, the anti PD-1/anti-PD-L1 therapies are characterized by an increased number of adverse effects, especially immune-mediated side effects. The increased activation of the immune system leads to systemic effects in a high number of cases, with the most frequent being represented by colitis, myocarditis, encephalitis, pneumonitis, hepatitis, etc. Fatal events have also been reported, but with a decreased incidence (0.3–1.3%). The frequency and the number of associated symptoms increase in the case of an association with two immune checkpoint inhibitors [97].

## 6. Conclusions

Although there are currently multiple clinical trials underway for immunotherapy, their results in ovarian cancer are not as spectacular as in melanoma or non-small-cell lung cancer, with this response being modulated by multiple internal factors. Identifying all these factors, as well as the interrelations between them and PD-1/PD-L1 is, thus, an important step, regarding understanding the therapeutic mechanisms of the disease. Succeeding research is necessary to certify the effectiveness of checkpoint immune inhibitors therapy in ovarian cancer, as well as to appraise the perspective role of anti-PDL-1 blockers, especially regarding the treatment of high grade malignant epithelial ovarian tumors, such as the clear-cell and mucinous carcinoma histological subtypes [56,82]. Currently there are many immune checkpoint inhibitors under study and development for cancer treatment. These new PD-1/PDL-1 inhibitors will need to overcome barriers, such as the lack of responsiveness in patients, adverse reactions, and drug resistance. In order to achieve these objectives, it is necessary to take the chemical structure of the inhibitors into account, as well as the future detection of biomarkers that indicate the selection path of patients who will not develop resistance to treatment.

## Figures and Tables

**Figure 1 ijms-23-12067-f001:**
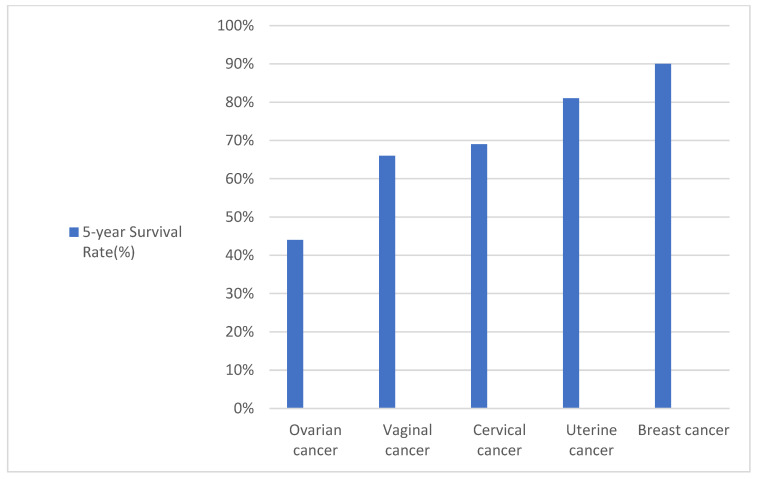
Histogram for 5-year survival rate in gynaecological cancers, regardless of the stage of the disease at the moment of presentation [3,4].

**Figure 2 ijms-23-12067-f002:**
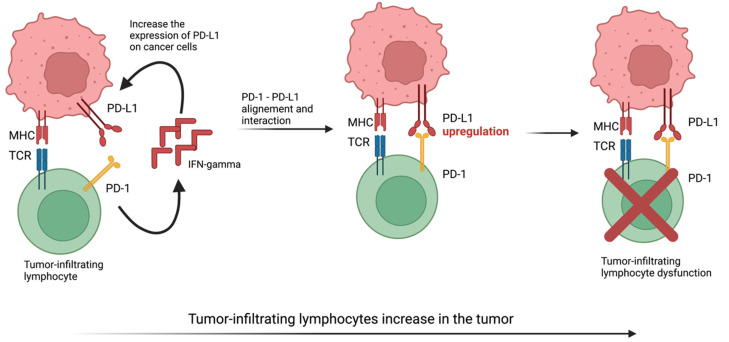
The interaction between PD-1 and PD-L1 leads to dysfunction of tumor infiltrating lymphocytes and evasion of tumor cells from the immune system. (TCR—T cel receptor, MHC-major histocompatibility complex). Created with BioRender.com (Last accessed on 26 September 2022).

**Figure 3 ijms-23-12067-f003:**
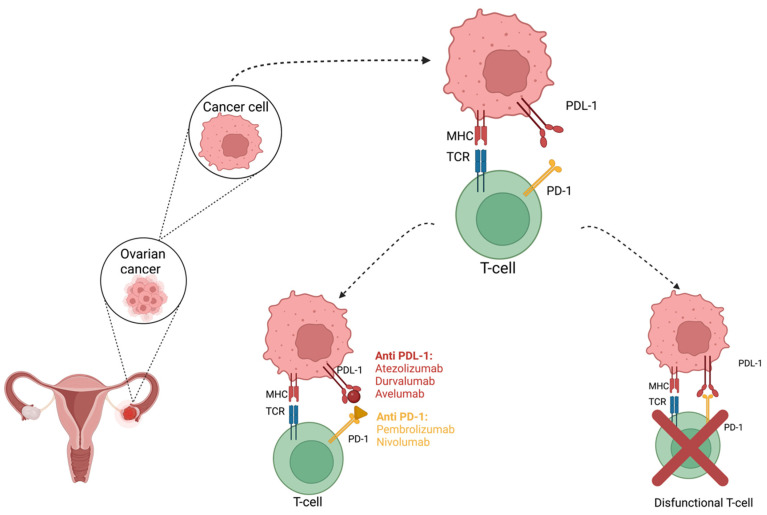
Antibody blockade of PD-1 and PD-L1. Anti-PD-1 and PD-L1 antibodies block the PD-1/PD-L1 signalling and enhance antitumor immune activity. Created with BioRender.com (Last accessed on 26 September 2022).

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
