# Peer review of "Focus on PD-1/PD-L1 as a Therapeutic Target in Ovarian Cancer"

_ijms, 2022, doi:10.3390/ijms232012067_

Round 1

Reviewer 1 Report

This interesting paper is reviewing current research advances in cancer treatment, with particular emphasis on the recent progress and future perspectives of the ovarian cancer immunotherapy, based on the PD-1/PD-L1 blockade. The focus has been placed on one of the most important molecules expressed on the surface of several immune cells, the programmed cell death receptor 1 (PD-1) and its ligands PD-L1 (B7-H1; CD274) and PD-L2.  The Authors have analyzed the most important immune checkpoint inhibitors including antibodies against the PD-1 receptor (nivolumab, pembrolizumab) as well as the PD-L1 ligand (avelumab, BMS - 936559, durvalumab, atezolizumab). The influence of microRNAs and long non-coding RNAs (lncRNA) on expression of PD-1/PD-L1 has also been delineated. This timely review addresses the growing interest in cancer treatment. I recommend the paper for publication after minor revision addressing the issues listed below.

1.      Recently, the high sensitivity of gated plasmonic biosensors for the determination of PD-L1 protein have been developed (e.g., Nanomaterials 2020 10:1592, doi: 10.3390/nano10081592). This relevant literature reference should be cited.

2.      There is no summarizing analysis of improvements in the effectiveness of immunotherapy. Some illustrations of the ovarian cancer progression and effectiveness of the immunotherapy would be beneficial for the Readers and would increase the citations.

Author Response

This interesting paper is reviewing current research advances in cancer treatment, with particular emphasis on the recent progress and future perspectives of the ovarian cancer immunotherapy, based on the PD-1/PD-L1 blockade. The focus has been placed on one of the most important molecules expressed on the surface of several immune cells, the programmed cell death receptor 1 (PD-1) and its ligands PD-L1 (B7-H1; CD274) and PD-L2.  The Authors have analyzed the most important immune checkpoint inhibitors including antibodies against the PD-1 receptor (nivolumab, pembrolizumab) as well as the PD-L1 ligand (avelumab, BMS - 936559, durvalumab, atezolizumab). The influence of microRNAs and long non-coding RNAs (lncRNA) on expression of PD-1/PD-L1 has also been delineated. This timely review addresses the growing interest in cancer treatment. I recommend the paper for publication after minor revision addressing the issues listed below.

We acknowledge #Reviewer 1 for the comments that definitely helped us to improve the manuscript.

  1. Recently, the high sensitivity of gated plasmonic biosensors for the determination of PD-L1 protein have been developed (e.g., Nanomaterials 2020 10:1592, plasmonic biosensors for the determination of PD-L1 protein have been developed (e.g., Nanomaterials 2020 10:1592, doi: 10.3390/nano10081592).). This relevant literature reference should be cited.

Thank you for this suggestion, we cited and discussed the suggested paper.

  1. There is no summarizing analysis of improvements in the effectiveness of immunotherapy. Some illustrations of the ovarian cancer progression and effectiveness of the immunotherapy would be beneficial for the Readers and would increase the citations.

Thank you for this suggestion, we all considered it very useful. We included two additional figures in the text.

Reviewer 2 Report

Focus on PD-1/PD-L1 as a therapeutic target in ovarian cancer

The authors present a comprehensive review of PD-1/PD-L1 as a therapeutic target in ovarian cancer.  

#1. I went through “iThenticate” to see whether there are some plagiarisms. It seems that the authors have gone through self-plagiarism (reference paper “Ref: Ro J Med Pract. 2021;16(Suppl7), the PDF file attached, the matching percentage is 26%).

#2. The role of immunotherapy has been emphasized in recent years, however their benefits to ovarian cancer have not yet been fully confirmed. In the aim, the authors wrote that they would “explain” the present status of the anti PD-1/PDL1 therapy in ovarian cancer. However, it is suggested that the authors review about the potential causes of the low efficacy of anti-PD-1/ PD-L1 in ovarian cancer and the present recent studies to overcome the low efficacy. It is not considered to be meaningful to “line up” the reference papers related to “ovarian cancer” and “PD-1/ PD-L1”.

Author Response

We acknowledge #Reviewer 2 for the pertinent observations.

The authors present a comprehensive review of PD-1/PD-L1 as a therapeutic target in ovarian cancer. 

#1. I went through “iThenticate” to see whether there are some plagiarisms. It seems that the authors have gone through self-plagiarism (reference paper “Ref: Ro J Med Pract. 2021;16(Suppl7), the PDF file attached, the matching percentage is 26%).

 Thank you for for doing this, we all are very grateful. Indeed, one of the authors took the paragraphs that he wrote and published them in other journal without notifying the co-authors. This is the reason for which we all discussed and agreed to remove him from our list of authors. See the attached documents signed in the system. All the paragraphs were re-written and some new ones were added to improve the manuscript.

#2. The role of immunotherapy has been emphasized in recent years, however their benefits to ovarian cancer have not yet been fully confirmed. In the aim, the authors wrote that they would “explain” the present status of the anti PD-1/PDL1 therapy in ovarian cancer. However, it is suggested that the authors review about the potential causes of the low efficacy of anti-PD-1/ PD-L1 in ovarian cancer and the present recent studies to overcome the low efficacy. It is not considered to be meaningful to “line up” the reference papers related to “ovarian cancer” and “PD-1/ PD-L1”.

We are thankful to the reviewer for the comments, we added several paragraphs about the efficacy of immune blockade inhibitors, however the data and the clinical studies are quite limited for the ovarian cancers. Our purpose was to review the literature focusing on the molecular mechanisms, not to explain the low efficacy of antibody blockade of PD-1 and PD-L1 in clinical trials.

Reviewer 3 Report

In this review article, the authors provided overview and current advance of immune checkpoint inhibitors in ovarian cancer treatment, summarized clinical studies of PD-1/PD-L1 inhibitors in ovarian cancer. This manuscript is well written, easy to follow and would be an important source in understanding the use of PD-1/PD-L1 inhibitors in ovarian cancer patients. 

Minor points:

1. Figure: please add y axis title in Figure 1.  

Additional figures and tables that summarize each section would be helpful. For example, a figure can be used to describe the molecular structures of PD-1 and PD-L1 in section 2. A table can be used to summarize all the clinical trials in ovarian cancer.

22.   Make sure the abbreviations are consistent, PD-1 and PD-L1 are not switched. For example, PD-1 (line 292, page 6) means PD-1, not PD-L1.

Author Response

We acknowledge #Reviewer 3 for the comments that definitely helped us to improve the manuscript.

In this review article, the authors provided overview and current advance of immune checkpoint inhibitors in ovarian cancer treatment, summarized clinical studies of PD-1/PD-L1 inhibitors in ovarian cancer. This manuscript is well written, easy to follow and would be an important source in understanding the use of PD-1/PD-L1 inhibitors in ovarian cancer patients.

Minor points:

  1. Figure: please add y axis title in Figure 1.

Thank you for this observation. We included the title 5-year Survival Rate (%)

Additional figures and tables that summarize each section would be helpful. For example, a figure can be used to describe the molecular structures of PD-1 and PD-L1 in section 2.

Thank you for this suggestion. We included 2 figures.

A table can be used to summarize all the clinical trials in ovarian cancer.

Thank you for this suggestion, we did not consider this useful since there are only few  active trials of checkpoint inhibitor in ovarian cancer and there is another paper analyzing these studies, please see reference 85.

  1. Make sure the abbreviations are consistent, PD-1 and PD-L1 are not switched. For example, PD-1 (line 292, page 6) means PD-1, not PD-L1.

Thank you for this observation. We corrected it.